# Hypomagnesemia Is Associated with the Acute Kidney Injury in Traumatic Brain Injury Patients: A Pilot Study

**DOI:** 10.3390/brainsci13040593

**Published:** 2023-03-31

**Authors:** Zhenjun Liu, Ruoran Wang, Min He, Yan Kang

**Affiliations:** 1Department of Critical Care Medicine, West China Hospital, Sichuan University, Chengdu 610041, China; 2Sichuan Cancer Hospital and Institute, Sichuan Cancer Center, School of Medicine, University of Electronic Science and Technology of China, Chengdu 610041, China; 3Department of Neurosurgery, West China Hospital, Sichuan University, Chengdu 610041, China

**Keywords:** acute kidney injury, magnesium, hypomagnesemia, traumatic brain injury, risk factor

## Abstract

Background: Acute kidney injury (AKI) commonly develops among traumatic brain injury (TBI) patients and causes poorer outcomes. We perform this study to explore the relationship between serum magnesium and the risk of AKI among TBI. Methods: TBI patients recorded in the Medical Information Mart for Intensive Care-III database were eligible for this research. The restricted cubic spline (RCS) was utilized to fit the correlation between serum magnesium level and the AKI. Univariate and subsequent multivariate logistic regression analysis were utilized to explore risk factors of AKI and confirmed the correlation between serum magnesium and AKI. Results: The incidence of AKI in included TBI was 21.0%. The RCS showed that the correlation between magnesium level and risk of AKI was U-shaped. Compared with patients whose magnesium level was between 1.5 and 2.0 mg/dL, those with a magnesium level of <1.5 mg/dL or >2.0 mg/dL had a higher incidence of AKI. Multivariate logistic regression confirmed age, chronic renal disease, ISS, serum creatinine, vasopressor, mechanical ventilation, and serum magnesium <1.5 mg/dL were independently related with the AKI in TBI. Conclusion: Abnormal low serum magnesium level is correlated with AKI development in TBI patients. Physicians should pay attention on renal function of TBI patients especially those with hypomagnesemia.

## 1. Introduction

As a kind of intracranial disease, TBI could cause a series of pathophysiological changes impairing function of extracranial organs. Previous studies have found that extracranial organ dysfunction was common among TBI patients with the incidence ranging from 33% to 89% [1,2,3,4]. As a kind of extracranial organ dysfunction, the acute kidney injury (AKI) has been noted as commonly developing after TBI with the incidence ranging from 9.2% to 24% [5,6,7,8]. Moreover, the AKI has been verified as being related to poorer prognosis of TBI victims [9,10]. To avoid the worsening renal function, many studies have been performed to explore risk factors for AKI in TBI patients and have confirmed some risk factors, including brain injury severity, high dose of hypertonic drugs, blood transfusion, and nephrotoxic antibiotics [11,12,13,14,15,16].

The magnesium is the fourth cation in the body, being second only to calcium, sodium, and potassium. It plays an important role in various activities, including neurotransmitter release, energy generation, cardiac movement, intracellular calcium regulation and protein metabolism [17,18]. The serum magnesium level would be influenced by several factors such as thyroid hormone, malnutrition, and gastrointestinal absorption. Additionally, previous studies have found that abnormal serum magnesium level was associated with the AKI development in several kinds of patients, including critically ill children, those diagnosed with acute pancreatitis, cancer, or undergoing cardiac surgery [19,20,21,22]. Despite this, there is still no study exploring the relationship between serum magnesium level and the AKI development in TBI patients. Therefore, this study was designed to verify the association between abnormal serum magnesium level and the risk of AKI after TBI.

## 2. Materials and Methods

### 2.1. Patients

Patients recorded in the Medical Information Mart for Intensive Care-III (MIMIC-III) database were eligible for this observational study. Collecting information from patients hospitalized in Beth Israel Deaconess Medical Center (BIDMC) (Boston, MA, USA) from 2001 to 2012, the free public MIMIC-III was developed by the computational physiology laboratory of Massachusetts Institute of Technology (MIT) (Cambridge, MA, USA). This database received approval from the institutional review boards of MIT and BIDMC. All patients included in the MIMIC-III were de-identified to protect their personal privacy. Patients with head trauma were included for this study from the MIMIC-III based on ICD-9 codes (80,000–80,199; 80,300–80,499; 8500–85,419). Patients were excluded from this study according to the following criteria: (1) Lacked records of vital signs and laboratory tests; (2) Lacked records of GCS on admission; (3) AKI developed within 24 h since admission (Figure 1). A total of 2470 patients were finally selected after screening.

### 2.2. Data Collection

Age, gender, and comorbidities, including diabetes, hypertension, hyperlipidemia, cerebral vascular disease, coronary heart disease, asthma, and cancer were collected. Initial vital signs, including systolic and diastolic blood pressure and heart rate were recorded. The Glasgow Coma Scale (GCS) on admission, abbreviated injury score (AIS) chest and Injury Severity Score (ISS) were also collected. The TBI severity was classified based on the GCS: mild (13–15); moderate (9–12); severe (3–8). Intracranial injury classifications, including subarachnoid hemorrhage, subdural hematoma, and epidural hematoma were identified from the radiological image. Results of laboratory tests from the first blood sample within the first day after admission were extracted, including white blood cell, red blood cell (RBC), platelet, hemoglobin, serum creatinine, blood urea nitrogen, serum potassium, serum sodium and serum magnesium. Medical treatments were collected, including RBC transfusion during the first 24 h, platelet transfusion during the first 24 h, vasopressor use within the first 24 h, neurosurgical operations, and mechanical ventilation. The main outcome of this study was the AKI diagnosed according to the KDIGO criteria 24 h after admission [23]. Other outcomes, including 30-day mortality, length of ICU stay, and length of hospital stay were collected and compared between groups of different magnesium levels. All variables were extracted by Structure Query Language using Navicat Premium 12. This research was performed complying with ethical standards of the Helsinki declaration.

### 2.3. Statistical Analysis

The normality of variables was verified utilizing Kolmogorov–Smirnov test. Variables of normal distribution and non-normal distribution were presented as mean ± standard deviation and median (interquartile range). Enumeration data were presented as numbers (percentage). Differences between groups of normal distribution and non-normal distribution variables were testified by ANOVA and the Kruskal–Wallis test, respectively. Chi-square test and Fisher exact test were performed to compare the difference between groups of enumeration data. The restricted cubic spline (RCS), one of the most common methods for analyzing nonlinear relationships, was used to fit the potential nonlinear correlation between serum magnesium level and the AKI risk among included TBI patients. The univariate logistic regression was first performed to explore the potential risk factors of AKI. Then the multivariate logistic regression was conducted to explore independent risk factors for AKI and verify the correlation between different magnesium levels and the AKI risk, adjusting significant factors in the univariate logistic regression.

Bilateral *p* value < 0.05 was considered statistically significant. SPSS 23.0 software (SPSS, Inc., Chicago, IL, USA) and R software (version 3.6.1; R Foundation, Indianapolis, IN, USA) were utilized to perform all statistical analyses and draw figures.

## 3. Results

### 3.1. Characteristics of TBI Patients Grouped by Serum Magnesium Level

A total of 2470 TBI patients were included with the AKI incidence of 21.0%. The RCS curve showed that the relationship between serum magnesium level and the risk of AKI was U-shaped (Figure 2). Therefore, patients were divided into three groups according to the magnesium level: group 1: <1.5 mg/dL (13.4%), group 2: 1.5–2.2 mg/dL (80.9%), group 3: >2.2 mg/dL (5.7%). The AKI incidence of these three groups were 28.2%, 19.4% and 27.7%, respectively (Figure 3). Age (*p* < 0.001), gender (*p* = 0.001), complicated incidence of diabetes (*p* = 0.001), hypertension (*p* = 0.001), coronary heart disease (*p* < 0.001), and chronic renal disease (*p* < 0.001) all differed between three groups (Table 1). Vital signs including systolic blood pressure (*p* < 0.001), diastolic blood pressure (*p* = 0.008), heart rate (*p* < 0.001), GCS (*p* < 0.001), and ISS (*p* < 0.001) were statistically different between three groups. Intracranial injury types including epidural hematoma (*p* = 0.010) and subdural hematoma (*p* = 0.008) were also different between the three groups. Laboratory tests showed white blood cells (*p* < 0.001), platelets (*p* = 0.033), red blood cells (*p* < 0.001), hemoglobin (*p* < 0.001), blood urea nitrogen (*p* < 0.001), serum creatinine (*p* < 0.001), potassium (*p* < 0.001), and chloride (*p* < 0.001) differed significantly between the three groups. Compared with group 3, group 1 and group 2 had higher usage incidence of RBC transfusion (*p* < 0.001) and platelet transfusion (*p* = 0.002). Group 1 and group 3 were more likely to receive vasopressor (*p* < 0.001) and neurosurgical operation than group 2 (*p* = 0.013). The percentage of AKI stage 1, 2, 3 in overall patients were 16.7%, 3.2% and 1.2%, respectively. Group 1 and group 3 had higher mortality (*p* = 0.002), longer length of ICU stay (*p* < 0.001), and length of hospital stay (*p* < 0.001) than group 2.

### 3.2. Distribution of Magnesium Group Classified by the TBI Severity

The distribution of magnesium groups classified by the TBI severity showed that group 1 was most likely to occur in severe TBI patients with the percentage of 20.2% (Table 2). The serum magnesium level was the lowest in severe TBI patients (*p* < 0.001).

### 3.3. Risk Factors of AKI in TBI Discovered by Logistic Regression

Univariate logistic regression presented that age (*p* < 0.001), diabetes (*p* < 0.001), coronary heart disease (*p* < 0.001), chronic renal disease (*p* < 0.001), GCS (*p* < 0.001), ISS (*p* < 0.001), platelets (*p* = 0.004), red blood cells (*p* < 0.001), hemoglobin (*p* < 0.001), blood urea nitrogen (*p* < 0.001), serum creatinine (*p* < 0.001) potassium (*p* = 0.023), magnesium <1.5 mg/dL (*p* < 0.001), magnesium > 2.2 mg/dL (*p* = 0.018), platelet transfusion (*p* < 0.001), use of vasopressor (*p* < 0.001), and mechanical ventilation (*p* < 0.001) were significantly associated with the AKI development (Table 3). While multivariate logistic regression showed age (*p* < 0.001), chronic renal disease (*p* = 0.010), ISS (*p* < 0.001), serum creatinine (*p* < 0.001), magnesium < 1.5 mg/dL (*p* = 0.015), use of vasopressor (*p* = 0.003), mechanical ventilation (*p* < 0.001) but not magnesium > 2.2 mg/dL (*p* = 0.905) were correlated with the AKI among TBI patients after adjusting for confounding effects.

## 4. Discussion

In this study, the magnesium < 1.5 mg/dL was confirmed as a risk factor for AKI among TBI patients. Group 1 contained 13.4% of TBI patients (Magnesium < 1.5 mg/dL) and 5.7% of them were divided into group 3 (Magnesium > 2.2 mg/dL). The widely recognized lower limit and upper limit of serum magnesium were 1.7 mg/dL and 2.7 mg/dL, respectively, which indicated that all patients of group 1 developed hypomagnesemia. The serum magnesium level is mainly influenced by several factors including thyroid hormone, malnutrition, renal function, and gastrointestinal absorption. The hypomagnesemia would be caused by primary hyperparathyroidism, excessive loss of digestive tract, excessive loss from the kidney and long-term insufficient nutrition supplement. As for TBI patients, the excessive loss of magnesium from the digestive tract may be a cause of the hypomagnesemia after TBI. As a common non-neurological complication after TBI, gastrointestinal dysfunction has been investigated, developing in 95% of TBI patients hospitalized in the intensive care unit [24,25,26]. Additionally, the excessive loss from the kidney by the massive use of diuretics mannitol and furosemide, which are commonly prescribed among TBI patients to reduce intracranial pressure, is another cause of the hypomagnesemia. These drugs could inhibit the reabsorption of magnesium from the medullary thick ascending or increase the excretion of magnesium from the kidney. The fact that serum magnesium level was the lowest in severe TBI patients in our study indicates patients with more severe injury may need more usage of these drugs. Of the TBI patients in this study, 5.7% were classified to group 3 (magnesium > 2.2 mg/dL) and only 0.6% (15/2470) developed hypermagnesemia. Actually, the hypermagnesemia is a rare biochemical abnormality which is mostly attributable to the usage of magnesium-containing drugs.

Several previous studies have explored the correlation between serum magnesium and the AKI risk in other kinds of patients including critically ill children, acute pancreatitis, cancer, and those undergoing cardiac surgery [19,20,21,22]. Three of them discovered that hypomagnesemia was independently correlated with the higher risk of AKI while another one found hypermagnesemia but not hypomagnesemia was correlated with the AKI in critically ill children [19,20,21,22]. Furthermore, one study with a large sample size verified that both hypermagnesemia and hypomagnesemia could increase the AKI risk in general hospitalized patients [27]. The divergence of correlation between serum magnesium level and AKI risk in these researches may arise from patients’ heterogeneity and different definition of abnormal serum magnesium level.

The underlying mechanism of the correlation between magnesium and AKI has not been definitely confirmed. There are several possible pathways to explain this association. Firstly, magnesium could enhance renal vasodilation by releasing endogenous vasodilators such as nitric oxide, adenosine, and prostaglandin and competitively inhibiting the calcium transport system [28,29,30,31]. Previous animal studies showed deficiency of magnesium would decrease renal blood flow, while supplements of magnesium could prevent the reduction of renal blood flow in post-ischemic AKI models [29,30,32]. Secondly, deficiency of magnesium could aggravate inflammation and oxidative stress through promoting the opening of calcium channels and secretion of proinflammatory cytokines, which are detrimental to renal function [33,34,35]. One study found that culture media with inadequate magnesium could lead to the increased production of proinflammatory transcription factors and cytokines expressed in endothelial cells [36]. As for hypermagnesemia, our study did not discover that high serum magnesium level was correlated with the AKI development in TBI patients. The divergence of the correlation between magnesium and AKI in several studies may originate from the pathophysiological heterogeneity or the different incidence and mechanism of hypermagnesemia in various diseases.

To sum up, physicians should try to maintain normal serum magnesium level and avoid the development of hypomagnesemia so as to decrease the risk of AKI in TBI patients. Supplementing adequate nutrition intake and decreasing the loss of magnesium from the gastrointestinal tract and the kidney may be beneficial to maintain the serum magnesium level. It remains to be verified by further trials whether recovering serum magnesium in TBI patients with hypomagnesemia could help reduce the risk of AKI.

Several limitations existed in the study. Firstly, this observational study was performed using data from a single medical center and therefore the selection bias may not be fully eliminated. Our conclusion should be confirmed by more studies conducted in other hospitals. Secondly, the casual relationship between magnesium level and the AKI risk is not enough to be proved by our research. Future prospective studies are valuable to explore the underlying mechanism between magnesium level and the AKI risk. Thirdly, we only included the serum magnesium level on admission. The serum magnesium level may fluctuate during hospitalization and the initial serum magnesium level may not fully reflect the effect of magnesium on AKI. A future study could be designed to dynamically collect magnesium levels so as to evaluate the effect of magnesium more objectively. Finally, this study was performed using data from a freely accessible database which did not record any organ specific biomarkers, including recombinant human ubiquitin carboxy terminal hydrolase isozyme L1 and glial fibrillary acid protein. Therefore, we could not include these markers in the analysis. Future prospective studies measuring these variables could be performed to verify our findings.

## 5. Conclusions

Abnormal low serum magnesium levels are correlated with a higher hazard of AKI among TBI patients. Physicians should pay more attention on preventing deteriorating renal function in TBI patients with hypomagnesemia.

## Figures and Tables

**Figure 1 brainsci-13-00593-f001:**
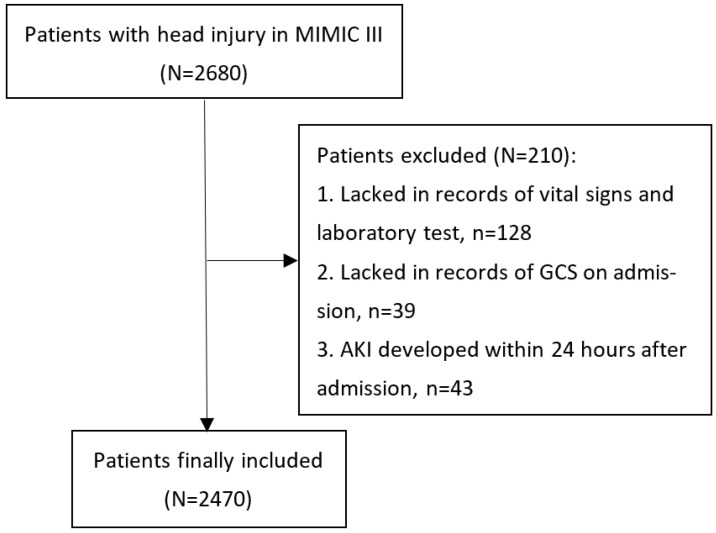
Flowchart of patients’ inclusion.

**Figure 2 brainsci-13-00593-f002:**
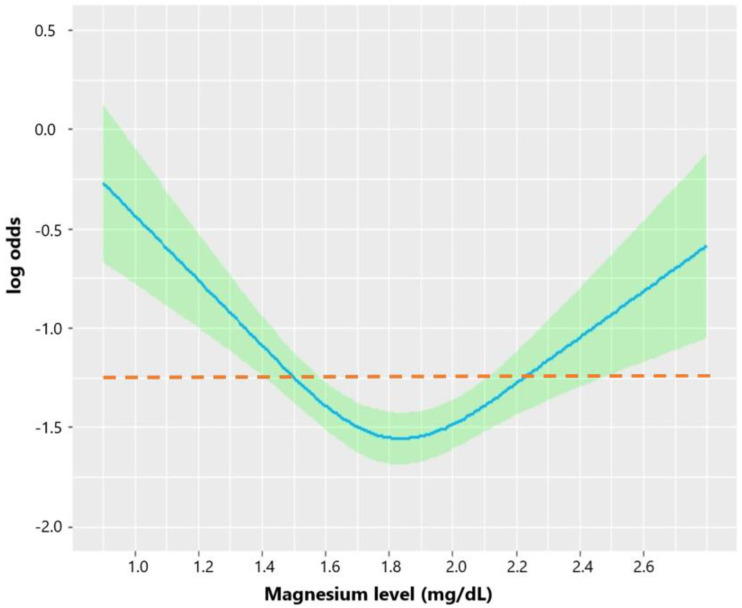
Association between magnesium level and the development of AKI analyzed by RCS.

**Figure 3 brainsci-13-00593-f003:**
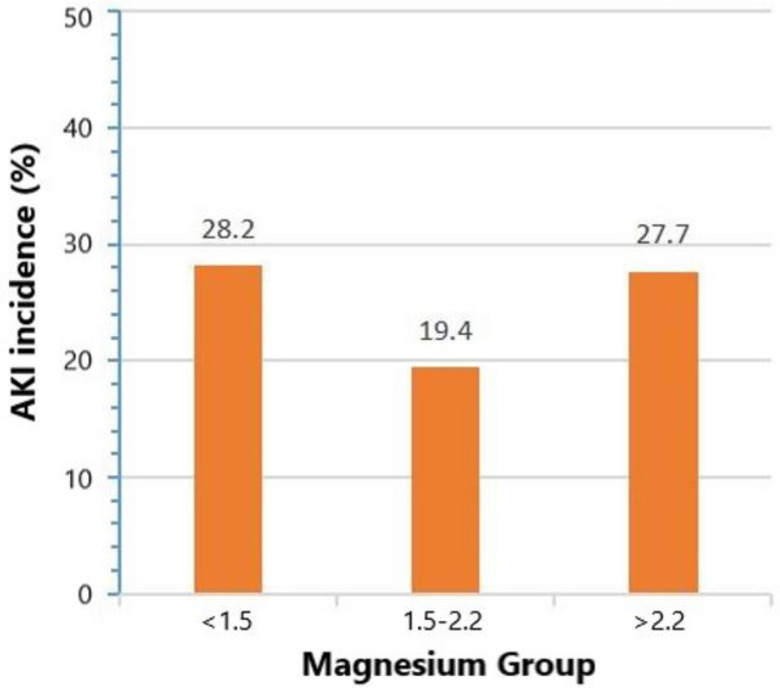
Incidence of AKI in groups of different magnesium levels.

**Table 1 brainsci-13-00593-t001:** Baseline characteristics of included TBI patients grouped by magnesium level.

Variables	Overall Patients (*n* = 2470)	Group 1: Magnesium < 1.5 mg/dL (*n* = 330, 13.4%)	Group 2: Magnesium 1.5–2.2 mg/dL (*n* = 1999, 80.9%)	Group 3: Magnesium > 2.2 mg/dL (*n* = 141, 5.7%)	*p*
Age (year)	62.0 (41.3–80.3)	50.6 (30.9–69.2)	62.9 (41.9–80.6)	78.5 (57.7–86.1)	<0.001
Gender (Male, %)	1519 (61.5%)	173 (52.4%)	1258 (62.9%)	88 (62.4%)	0.001
Comorbidities					
Diabetes (%)	352 (14.3%)	44 (13.3%)	273 (13.7%)	35 (24.8%)	0.001
Hypertension (%)	872 (35.3%)	86 (26.1%)	737 (36.9%)	49 (34.8%)	0.001
Cerebral vascular disease (%)	44 (1.8%)	3 (0.9%)	41 (2.1%)	0	0.090
Coronary heart disease (%)	297 (12.0%)	12 (3.6%)	257 (12.9%)	28 (19.9%)	<0.001
Chronic renal disease (%)	145 (5.9%)	10 (3.0%)	101 (5.1%)	34 (24.1%)	<0.001
Vital signs on admission					
Systolic blood pressure (mmHg)	132 (117–147)	127 (108–141)	132 (118–147)	132 (118–147)	<0.001
Diastolic blood pressure (mmHg)	67 (56–77)	65 (54–75)	67 (57–78)	65 (54–77)	0.008
Heart rate (s^−1^)	84 (72–96)	90 (77–103)	83 (72–95)	82 (71–91)	<0.001
Respiratory rate (s^−1^)	18 (15–20)	18 (15–20)	18 (15–20)	17 (15–20)	0.614
GCS	11 (6–15)	8 (3–12)	13 (7–15)	14 (8–15)	<0.001
TBI severity					<0.001
Mild (%)	1197 (48.5%)	80 (24.2%)	1035 (51.8%)	82 (58.2%)	
Moderate (%)	338 (13.7%)	61 (18.5%)	260 (13.0%)	17 (12.1%)	
Severe (%)	935 (37.9%)	189 (57.3%)	704 (35.2%)	42 (29.8%)	
ISS	16 (16–25)	20 (16–29)	16 (16–24)	16 (16–18)	<0.001
Intracranial injury type					
Epidural hematoma (%)	547 (22.1%)	84 (25.5%)	445 (22.3%)	18 (12.8%)	0.010
Subdural hematoma (%)	1316 (53.3%)	159 (48.2%)	1067 (53.4%)	90 (63.8%)	0.008
Subarachnoid hemorrhage (%)	957 (38.7%)	132 (40.0%)	782 (39.1%)	43 (30.5%)	0.112
Intraparenchymal hemorrhage (%)	445 (18.0%)	55 (16.7%)	371 (18.6%)	19 (13.5%)	0.250
Laboratory tests					
White blood cell (10^9^/L)	11.80 (8.60–15.90)	13.45 (8.80–18.17)	11.60 (8.60–15.60)	10.40 (7.50–15.70)	<0.001
Platelet (10^9^/L)	233 (185–288)	219.50 (170–287)	235 (189–288)	234 (177–285)	0.033
Red blood cell (10^9^/L)	4.16 (3.69–4.60)	3.81 (3.39–4.31)	4.20 (3.75–4.63)	4.18 (3.74–4.53)	<0.001
Hemoglobin (g/dL)	12.9 (11.5–14.2)	11.9 (10.6–13.4)	13.0 (11.7–14.3)	12.7 (11.1–14.1)	<0.001
Blood urea nitrogen (mg/dL)	16 (12–22)	14 (10–19)	16 (12–22)	24 (15–45)	<0.001
Serum creatinine (mg/dL)	0.9 (0.7–1.1)	0.8 (0.7–1.1)	0.9 (0.8–1.1)	1.1 (0.9–1.7)	<0.001
Sodium (mmol/L)	139 (137–141)	140 (137–142)	139 (137–141)	139 (136–141)	0.624
Potassium (mmol/L)	4.0 (3.7–4.3)	3.8 (3.5–4.1)	4.0 (3.7–4.3)	4.3 (3.9–4.7)	<0.001
Chloride (mmol/L)	104 (101–107)	106 (102–110)	104 (101–107)	102 (99–106)	<0.001
Medical interventions					
RBC transfusion (%)	202 (8.2%)	59 (17.9%)	137 (6.9%)	6 (4.3%)	<0.001
Platelet transfusion (%)	222 (9.0%)	46 (13.9%)	167 (8.4%)	9 (6.4%)	0.002
Vasopressor use (%)	151 (6.1%)	44 (13.3%)	96 (4.8%)	11 (7.8%)	<0.001
Mechanical ventilation (%)	1137 (46.0%)	222 (67.3%)	872 (43.6%)	43 (30.5%)	<0.001
Neurosurgical operation (%)	580 (23.5%)	98 (29.7%)	447 (22.4%)	35 (24.8%)	0.013
AKI stage (%)					0.002
None	1951 (79.0%)	237 (71.8%)	1612 (80.6%)	102 (72.3%)	
1	412 (16.7%)	69 (20.9%)	310 (15.5%)	33 (23.4%)	
2	78 (3.2%)	16 (4.8%)	58 (2.9%)	4 (2.8%)	
3	29 (1.2%)	8 (2.4%)	19 (1.0%)	2 (1.4%)	
30-day mortality (%)	406 (16.4%)	64 (19.4%)	306 (15.3%)	36 (25.5%)	0.002
Length of ICU stay (day)	2.3 (1.2–5.5)	3.7 (1.8–9.9)	2.1 (1.2–5.0)	2.3 (1.6–4.5)	<0.001
Length of hospital stay (day)	6.4 (3.6–12.2)	9.7 (4.7–17.7)	5.9 (3.5–11.5)	6.5 (3.7–10.6)	<0.001

GCS, Glasgow Coma Scale; ISS, Injury Severity Score; RBC, red blood cell.

**Table 2 brainsci-13-00593-t002:** Distribution of magnesium groups classified by the TBI severity.

	Overall Patients (*n* = 2470)	Mild TBI (*n* = 1197, 48.5%)	Moderate TBI (*n* = 338, 13.7%)	Severe TBI (*n* = 935, 37.9%)	*p*
Magnesium group					<0.001
Group 1: Magnesium <1.5 mg/dL	330 (13.4%)	80 (6.7%)	61 (18.0%)	189 (20.2%)	
Group 2: Magnesium 1.5–2.2 mg/dL	1999 (80.9%)	1035 (86.5%)	260 (76.9%)	704 (75.3%)	
Group 3: Magnesium >2.2 mg/dL	141 (5.7%)	82 (6.9%)	17 (5.0%)	42 (4.5%)	
Serum magnesium level (mg/dL)	1.8 (1.6–2.0)	1.9 (1.7–2.1)	1.7 (1.5–2.0)	1.7 (1.5–1.9)	<0.001

**Table 3 brainsci-13-00593-t003:** Risk factors of AKI analyzed by univariate and multivariate logistic regression.

Variables	Univariate Logistic Regression Analysis				Multivariate Logistic Regression Analysis		
	OR	95% CI	*p*		OR	95% CI	*p*
Age	1.014	1.009–1.018	<0.001		1.015	1.009–1.021	**<0.001**
Male gender	0.939	0.770–1.144	0.531				
Comorbidities							
Diabetes	1.899	1.479–2.438	<0.001		1.163	0.870–1.556	0.308
Hypertension	1.052	0.860–1.287	0.622				
Cerebral vascular disease	1.259	0.632–2.508	0.513				
Coronary heart disease	1.801	1.377–2.356	<0.001		1.252	0.920–1.704	0.153
Chronic renal disease	4.272	3.036–6.010	<0.001		1.795	1.149–2.805	**0.010**
Vital signs on admission							
Systolic blood pressure	1.000	0.996–1.004	0.928				
Diastolic blood pressure	0.995	0.989–1.001	0.082				
Heart rate	1.003	0.997–1.008	0.317				
Respiratory rate	1.012	0.995–1.030	0.156				
GCS	0.942	0.922–0.962	<0.001		1.009	0.977–1.042	0.590
ISS	1.027	1.016–1.038	<0.001		1.024	1.012–1.037	**<0.001**
Intracranial injury type							
Epidural hematoma	0.905	0.714–1.147	0.409				
Subdural hematoma	0.985	0.812–1.196	0.880				
Subarachnoid hemorrhage	0.939	0.769–1.147	0.537				
Intraparenchymal hemorrhage	0.896	0.693–1.159	0.403				
Laboratory tests							
White blood cell	1.001	0.998–1.005	0.457				
Platelet	0.998	0.997–0.999	0.004		1.000	0.999–1.001	0.938
Red blood cell	0.613	0.534–0.705	<0.001		0.887	0.636–1.236	0.479
Hemoglobin	0.843	0.805–0.884	<0.001		0.989	0.884–1.107	0.850
Blood urea nitrogen	1.035	1.027–1.044	<0.001		1.006	0.995–1.018	0.288
Serum creatinine	2.132	1.772–2.566	<0.001		1.534	1.237–1.903	**<0.001**
Sodium	0.996	0.978–1.014	0.657				
Potassium	1.169	1.022–1.339	0.023		1.018	0.871–1.189	0.822
Chloride	0.998	0.982–1.014	0.787				
Magnesium							
1.5–2.2	1.000	Reference			1.000	Reference	
<1.5	1.635	1.255–2.129	<0.001		1.446	1.074–1.949	**0.015**
>2.2	1.593	1.083–2.341	0.018		0.972	0.610–1.548	0.905
Medical interventions							
RBC transfusion	1.299	0.931–1.813	0.124				
Platelet transfusion	1.845	1.365–2.494	<0.001		1.238	0.882–1.736	0.217
Vasopressor	2.754	1.959–3.872	<0.001		1.769	1.212–2.581	**0.003**
Mechanical ventilation	2.302	1.887–2.809	<0.001		2.520	1.885–3.368	**<0.001**
Neurosurgery	1.190	0.952–1.487	0.126				

OR, odds ratio; CI, confidence interval; GCS, Glasgow Coma Scale; ISS, Injury Severity Score; RBC, red blood cell.

## Data Availability

The datasets used for the current study are available from the corresponding author on reasonable request.

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
