# Peer review of "Hypomagnesemia Is Associated with the Acute Kidney Injury in Traumatic Brain Injury Patients: A Pilot Study"

_brainsci, 2023, doi:10.3390/brainsci13040593_

Round 1
Reviewer 1 Report
The paper entitled " Hypomagnesemia is associated with the acute kidney injury in traumatic brain injury patients" reports the relationship between serum magnesium and AKI in TBI patients. By analyzing more than 2000 patients, the authors reported that abnormal low serum magnesium could be related to the development of AKI in TBI patients. Therefore, authors proposed that the serum magnesium should be paid attention by physicians.
Overall the data supported the conclusion and the sample size is convincing. However, I would like to see a more detailed description about the specific method you used to analyze different set of data either in figure legends or result sections. Even though, you proposed the fact that serum magnesium is related to AKI, I don't think you gave a proper discussion about the possible mechanisms behind this phenomenon in the discussion section. I would like to see a improved discussion section. Also, there are some typos and confusing descriptions in the paper. Please fix them during the revision.
Author Response
Response: Thanks for your suggestion. The more detailed description about the statistical method we used has been added in the method part of our revised manuscript as the following: “The restricted cubic spline (RCS), one of the most common methods for analyzing nonlinear relationships, was used to fit the potential nonlinear correlation between serum magnesium level and the AKI risk among included TBI patients. The univariate logistic regression was firstly performed to explore potential risk factors of AKI. Then the multivariate logistic regression was conducted to explore independent risk factors for AKI and verify the correlation between different magnesium levels and the AKI risk adjusting significant factors in the univariate logistic regression.” We proposed the fact that serum magnesium is related to AKI. And we added one table in the revised manuscript showing the distribution of magnesium group classified by the TBI severity. This table (table 2) showed the group 1 (Magnesium <1.5 mg/dL) was most likely to occur in severe TBI patients with the percentage of 20.2%. And the serum magnesium level was the lowest in severe TBI patients. Therefore, the logic thread of the discussion is that TBI affecting magnesium levels which in turn increase the risk of AKI after TBI. The discussion section has been improved in our revised manuscript. And the English has been revised in our revision.
Reviewer 2 Report
We read with interest the article by Liu et al titled “Hypomagnesemia is associated with acute kidney injury (AKI) in traumatic brain injury patients”. The authors aim to explore the relationship between serum magnesium and the risk of AKI among TBI.
The work is more descriptive rather than mechanistically driven or at least not being hypothesis-driven. Patient assessment is showing a rough correlation between hypomagnesium levels and the development of AKI. The study suffers from a number of limitations which reflect on the data being preliminary.
Comments:
What is not clear from this study is the relation between TBI-AKI and hypo magnesium levels; ie what is driving what.
Is TBI affecting magnesium levels that would in turn lead to AKI or are TBI and AKI cooccurring and they are affected by the low magnesium levels? This should be clarified in the write-up as the design of this requires some modification:
1-First the TBI patients should be stratified according to severity: mild, moderate, and severe.
2- Similarly the AKI should be categorized into the mild and severe kidney injury
3- the measurement of magnesium should be mentioned if it is taken at similar time points among the patients and also the age of these TBI patients should be matched according to different category brackets.
4- the study lacked any biomarker data for both the TBI and AKI which are organ specific such as UCH-L1 and GFAP etc.
5- based on limitation, I think this work should fall in the category of: Pilot Study and the title thus should be : “Hypomagnesemia is associated with acute kidney injury (AKI) in traumatic brain injury patients: A Pilot Study
Minor Comments:
English Editing is required: the introduction needs to be checked for English wording
Author Response
Comments:
What is not clear from this study is the relation between TBI-AKI and hypo magnesium levels; ie what is driving what.
Is TBI affecting magnesium levels that would in turn lead to AKI or are TBI and AKI cooccurring and they are affected by the low magnesium levels? This should be clarified in the write-up as the design of this requires some modification:
Response: Thanks for this valuable advice. Actually, the hypothesis of our study is that TBI affecting magnesium levels which in turn increase the risk of AKI after TBI. The main outcome of this study was the AKI diagnosed according to the KDIGO criteria 24 hours after admission. The measurement timepoint of magnesium has been mentioned in the method part of our revised manuscript as the following: “Results of laboratory test from the first blood sample within the first day after admission”. Therefore, the chronological sequence between measurement timepoint of magnesium (within the first 24h) and the AKI diagnosis (after the first 24h) indicated the casual relationship between the serum magnesium level (the cause) and the risk of AKI (the result). Certainly, the casual relationship between magnesium level and the AKI risk should be further verified in prospective studies and even interventional trials. This limitation has been stated in the last paragraph of our manuscript. Additionally, we added one table in the revised manuscript showing the distribution of magnesium group classified by the TBI severity. This table (table 2) showed the group 1 (Magnesium <1.5 mg/dL) was most likely to occur in severe TBI patients with the percentage of 20.2%. And the serum magnesium level was the lowest in severe TBI patients.
- First the TBI patients should be stratified according to severity: mild, moderate, and severe.
Response: Thanks for this suggestion. We have stratified TBI patients to mild, moderate, and severe according to the GCS. This result has been updated in the revised table and manuscript. The distribution of magnesium group classified by the TBI severity was showed as the table 2 of our revised manuscript.
- Similarly the AKI should be categorized into the mild and severe kidney injury
Response: Thanks for this suggestion. The AKI in our manuscript has been categorized to stage 1, stage 2, stage 3 according to the KDIGO criteria. Did you mean categorize the stage 1 and stage 2 to mild and stage 3 to severe AKI. If you think this categorization is necessary, we would revise it in the next revision.
3- the measurement of magnesium should be mentioned if it is taken at similar time points among the patients and also the age of these TBI patients should be matched according to different category brackets.
Response: Thanks for this advice. The measurement timepoint of magnesium has been mentioned in the method part of our revised manuscript as the following: Results of laboratory test from the first blood sample within the first day after admission were extracted including white blood cell, red blood cell (RBC), platelet, hemoglobin, serum creatinine, blood urea nitrogen, serum potassium, serum sodium and serum magnesium. Certainly, the variation of timepoint measuring magnesium could not be fully avoided due to the limitation of the observational study. While this variation is restricted within the first 24 hour after admission and the timepoint of the first blood sample is usually relatively regular in the single medical center. We did not match the age according to different magnesium level category. Did you mean the age is a confounding factor of the association between serum magnesium level and the risk of AKI after TBI. While the multivariate logistic regression has been used in our study to adjusting confounding effects. If you consider the age as a confounding factor, while other potential confounding factors should also be matched according to different magnesium level category.
4- the study lacked any biomarker data for both the TBI and AKI which are organ specific such as UCH-L1 and GFAP etc.
Response: Thanks for this comment. This is a limitation of our study which we have stated in the limitation part of our revised manuscript. This study was performed using data from a freely accessible database. This database did not record any organ specific biomarker including UCH-L1 and GFAP. Therefore, we could not include these markers into the analysis. Future prospective studies measuring these variables could be performed to verify our findings.
5- based on limitation, I think this work should fall in the category of: Pilot Study and the title thus should be : “Hypomagnesemia is associated with acute kidney injury (AKI) in traumatic brain injury patients: A Pilot Study
Response: Thanks for this valuable suggestion. We have revised the title as you recommended.
Minor Comments:
English Editing is required: the introduction needs to be checked for English wording
Response: Thanks for this suggestion. We have invited an English professor revising our manuscript. The English wording of the whole manuscript including introduction has been revised.
Round 2
Reviewer 2 Report
Accept